# Galectin-3 did not associate with malaria-related insulin resistance in diabetic and non-diabetic respondents at a Ghanaian General Hospital

Emmanuel Nortey[1], Leonard Derkyi-Kwarteng[2], Daniel Amoako-Sakyi[1], Ansumana Sandy Bockarie[3], Samuel Yeboah[4], Samuel Acquah[5]*

1 Department of Microbiology and Immunology, School of Medical Sciences, College of Health and Allied Sciences, University of Cape Coast, Cape Coast, Ghana, 2 Department of Pathology, School of Medical Sciences, College of Health and Allied Sciences, University of Cape Coast, Cape Coast, Ghana, 3 Department of Internal Medicine and Therapeutics, School of Medical Sciences, College of Health and Allied Sciences, University of Cape Coast, Cape Coast, Ghana, 4 Ghana Ports and Harbours Authority Clinic, Tema, Ghana, 5 Department of Medical Biochemistry, School of Medical Sciences, College of Health and Allied Sciences, University of Cape Coast, Cape Coast, Ghana

* sacquah@ucc.edu.gh

## Abstract

### Background

Malaria remains endemic in the sub-Saharan African region. The region also faces the world's highest increase in incidence of type 2 diabetes mellitus (T2DM). Although galectin-3 has been explored in numerous conditions, scientific information on the relationship between malaria-related insulin resistance and circulating galectin-3 levels is limited. Therefore, the current study examined the association between galectin-3 and insulin resistance in diabetic and non-diabetic adults with or without malaria at the Tema General Hospital.

### Methods

Anthropometric indices, blood pressure, glucose, full blood count (FBC), lipid profile, insulin and galectin-3 levels were measured under fasting conditions. Insulin resistance and beta-cell function were assessed using the homeostatic model assessment of insulin resistance (HOMA-IR) and beta-cell function (HOMA-B) formulae.

### Results

Participants with T2DM were older (P < 0.05) with higher levels of systolic blood pressure and glucose but lower parasite levels than their non-diabetic counterparts. Irrespective of diabetes status, levels of total cholesterol, low-density lipoprotein cholesterol, high-density lipoprotein cholesterol and galectin-3 were higher but triglyceride level was lower in participants with malaria. Levels of insulin, HOMA-B and HOMA-IR were highest for diabetics without malaria with high strengths of the

**Data availability statement:** All relevant data are provided within the paper.

**Funding:** The author(s) received no specific funding for this work.

**Competing interests:** The authors declare that they have is no conflict of interest.

associations. Galectin-3 could neither predict HOMA-B nor HOMA-IR in any of the study groups. Irrespective of malaria or diabetes status, insulin resistance associated with glucose ($B = 0.603$, *Wald* = 10.52, *Exp (B)* = 1.83, CI: 1.27–2.63; P = 0.001) and insulin ($B = 1.145$, *Wald* = 18.61, *Exp (B)* = 3.14, CI: 1.87–5.23; P < 0.001) levels in our context with the model explaining 67.7% (Cox & Snell $R^2 = 0.677$) to 91% (Nagelkerke $R^2 = 0.91$) of the observed variation.

## Conclusion

The relationship of galectin-3 with HOMA-IR and HOMA-B appears more complex than a linear fashion in our setting.

---

## Introduction

Globally, the sub-Saharan African region bears 93.6% and 95.4% of all malarial cases and deaths respectively [1] making the condition mainly an African problem. In addition, the region has the lowest prevalence of diabetes at 4.5%, but the highest proportion of undiagnosed cases at 53.6% [2]. Above all, the sub-Saharan African region is predicted to have the highest increase in incidence of 129% and impaired glucose tolerance of 105% by 2045 [2]. These estimates point to the need to investigate the impact of the interaction between malaria and diabetes on the health and wellbeing of the affected. Indeed, malaria is associated with insulin resistance [3], an important risk factor for the development of type 2 diabetes mellitus (T2DM), which accounts for more than 90% of global diabetes cases [2]. Apart from malaria, other infectious agents have been linked to insulin resistance, suggesting that, infection in general, may be a probable risk factor for T2DM [3–5].

Galectin-3 is a chimeric vertebrate protein of 250 amino acids long [6]. It has a distinct C-terminal carbohydrate recognition domain and N-terminal protein-binding domain [7] and expressed by several tissues, including the gut, spleen, colon, kidney, and inflammatory cells such as mast cells, neutrophils and macrophages [6]. Galectin-3 is involved in various conditions such as diabetes, inflammation, fibrosis, rheumatoid arthritis, asthma, certain cancers and heart failure [8–11]. Compared to healthy controls, persons with T2DM have higher levels of galectin-3 [12] although an earlier study associated low levels with diabetes [13], suggesting the need for further studies to define the exact role of this glycoprotein in the development of diabetes mellitus. At the molecular level, overexpression of galectin-3 in skeletal muscle inhibits insulin signaling and glucose uptake [14,15]. With respect to malaria, galectin-3 was postulated to promote cerebral malaria in experimental settings as galectin-3 deficient mice was protected [16]. However, circulating level of galectin-9 is thought to reflect severity of malaria in humans [17,18]. In Ghana, adult malaria is normally mild to moderate in nature because such individuals have some level of immunity against the parasite due probably to previous exposures [3]. Moreover, improved public education on the signs and symptoms of malaria in our setting means that the adult population is unlikely to delay in seeking the needed treatment, thus, preventing

severe illness. However, scientific information on the role of galectin-3 in malaria-related insulin resistance in diabetic and non-diabetic adults is limited. Access to this information may deepen our understanding of malaria-related insulin resistance and contribute to efforts at reducing the menace of T2DM and its complications in the Ghanaian context. Therefore, this cross-sectional study was designed to examine the linear relationship between galectin-3 and malaria-related insulin resistance in diabetic and non-diabetic Ghanaians receiving treatment at the Tema General Hospital (TGH).

## Materials and methods

### Study site, design, population and sample size estimation

This cross-sectional study was conducted at the TGH, which serves as a secondary facility responsible for the healthcare needs of residents of Tema and its surrounding communities between January, 2023 and December, 2023. Participants were adults of both gender, aged 20 years and above, with or without malaria, without a history of COVID-19 but vaccinated against the condition. Pregnant and nursing mothers, and individuals with hepatitis, human immunodeficiency virus (HIV), pancreatitis, bacterial infections, rheumatoid arthritis, asthma, heart failure, T1DM, diabetes insipidus, liver and kidney diseases were excluded as these conditions can influence the measured biomarkers. Using the formula, $N = \left\{ P1(1-P1) + P2(1-P2) \right\} \left( \frac{Z\alpha + Z\beta}{P1-P2} \right)^2$, where the prevalence of malaria (P1) was estimated to be 16% [1], with that of diabetes (P2) at 1.8% [2], a $Z_\alpha$ of 1.96 for a two-tailed test and a $Z_\beta$ at 0.84 for 80% power, the sample size per group was estimated to be 59 respondents. Therefore, a total sample should have been 118 respondents. With an assumed non-response rate of 35%, 160 respondents were eventually used for the current study, made up of 80 diabetic and an equal number of non-diabetic respondents. Participants' recruitment for the study started on Monday, 16th January, 2023 and ended on Friday, 15th December, 2023.

### Sample selection

The simple random sampling technique was employed in selecting participants from the out-patient department and diabetes clinic of the TGH on a daily basis until the required number of participants was obtained (Fig 1).

### Demographic information, blood pressure and anthropometry

A developed questionnaire was used for collection of demographic and other lifestyle information including age, gender, anthropometry, exercise and medication intake.

An experienced nurse measured the blood pressure with an electronic sphygmomanometer (Mindray VS-900C, Salvin Dental Specialties, Inc, USA). The average of three measurements was reported as the blood pressure for each participant. Body weight, height, waist and hip circumferences were measured following the World Health Organization (WHO)

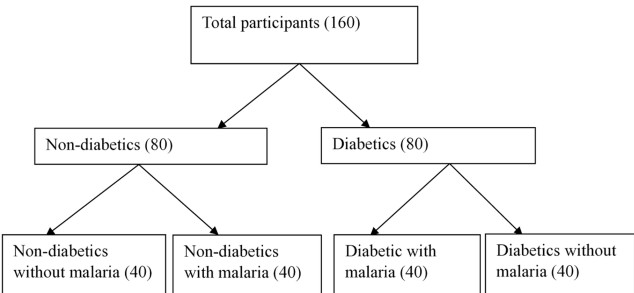

**Fig 1. Flow chart of participants' groupings.**

guidelines [19]. Body mass index (BMI) was computed as the ratio of body weight (kg) to the square of height (m²) but the waist-to-hip ratio (WHR) was obtained by dividing the waist circumference by the hip circumference. The body weight was measured to the nearest 0.1 kg using an electronic scale (Omron® BHF-510, Il, USA) with participants in light clothing. Height was measured to the nearest 0.1 cm using a stadiometer (Seca® mechanical column scale with stadiometer, USA). Waist and hip circumferences were measured to the nearest 0.1 cm with a flexible tape measure.

## Measurement of biochemical indices

Seven millilitres of venous blood sample was taken from each participant after an overnight fast and aliquoted into serum separator tube (4 ml), and ethylenediaminetetraacetic acid (EDTA) anticoagulant tube (3 ml). Serum samples were used for the measurement of lipid profile including total cholesterol, low-density lipoprotein cholesterol, high-density lipoprotein cholesterol and triglycerides, by the AU 480 Beckman coulter chemistry analyser (Beckman coulter, Ireland Inc., Ireland). However, glucose was measured from plasma sample using the same chemistry analyser following the glucose oxidase principle.

With respect to serum insulin level, a commercially available sandwich enzyme-linked immunosorbent assay (ELISA) kit for human insulin procured from PerkinElmer (PerkinElmer Health Sciences Inc., USA) was used for the measurement, following the manufacturers' instructions. A 96-well microplate pre-coated with anti-insulin antibody was used. Samples, standards, and reagents were brought to room temperature prior to the tests. Fifty microliters each of the various concentrations of insulin standards, controls and specimens was placed in appropriate wells. Exactly 100 $\mu$l of enzyme conjugate was added to each well. After a complete mixing process, the setup was then incubated for 1 hour. Employing an automated microplate washer (Thermo Electron Co-operation, Finland), each of the wells in the microplate was washed five rounds, with 300 $\mu$l of 1X wash buffer. Hundred microlitres of 3,3′,5,5′-tetramethylbenzidine (TMB) substrate was then added to each well for colour development, followed by incubation at room temperature in the dark for 20 minutes. Afterwards, the reaction was stopped by adding 100 $\mu$l of stop solution to each well, and thoroughly mixed until the blue color turned yellow. Using a multiscan microplate reader, the absorbance at 450 nm was promptly measured. A suitable standard curve was prepared using the absorbance values obtained from the insulin standards against its concentration. The insulin concentration in the sample was then determined from the standard curve. The sensitivity of the assay kit used for the measurement was 2 $\mu$IU/l.

The homeostatic model assessment of insulin resistance (HOMA-IR), $HOMA-IR = Glucose\ \left(\frac{mmol}{l}\right) * Insulin\ (m\frac{U}{l})/22.5$ and beta-cell function, $HOMAB = 20 * Fasting\ Insulin\ (\frac{mU}{l})/\ Fasting\ Glucose\ (\frac{mmol}{l}) - 3.5$ [20] formulae were employed for assessing insulin resistance and beta-cell function respectively. Insulin resistance was determined at HOMA-IR > 2.6 [21].

Serum levels of galectin-3 were measured by human galectin-3 quantikine ELISA Kit (R&D Systems Inc., USA) following manufacturers' protocol. The assay makes use of a quantitative sandwich ELISA technique in which a monoclonal antibody specific for human galectin-3 is pre-coated onto a 96-well microplate. Exactly 100 $\mu$l of assay diluent was added to each well followed by 50 $\mu$l of the various concentrations of standards, controls and samples in appropriate wells at room temperature, covered and incubated for two hours. Each well was then washed four times each with 400 $\mu$l of wash buffer using a multiwell plate washer (Sigma-Aldrich, Merck M2656, Germany). The plate was inverted and blotted against clean paper towels. Exactly 200 $\mu$l of human galectin-3 conjugate was then added to each well and incubated at room temperature for another two hours. The set up was washed as described earlier before the addition of 200 $\mu$L of substrate solution to each well followed by 30 minutes incubation at room temperature and away from light. Exactly 50 $\mu$L of stop solution was then added to each well until the color changed from blue to yellow. With a fully automated microplate reader (PKL PPC 142, Italy), the absorbance was then read at 450 nm and 570 nm. The difference in absorbance at 450 nm and 570 nm, which represent the corrected absorbance value, was used in determining the concentration of the galectin-3. The corrected absorbance for the standard galectin-3 was plotted on the y-axis against concentration of

galectin-3 to prepare a standard curve. The concentration of galectin-3 in sample was then determined from the standard curve in accordance with manufacturer's instructions. The sensitivity of this assay kit was 0.016 ng/ml.

### Full blood count

An automated haematology analyser (XN 1000, Sysmex Corporation, Japan) was used to determine the full blood count parameters. The specific parameters were haemoglobin, neutrophil, eosinophil, basophil, haematocrit, lymphocyte, platelet, red blood cell, white blood cell, mean corpuscular volume, mean corpuscular haemoglobin and mean corpuscular haemoglobin concentration.

### Malaria diagnosis

Malaria diagnosis was done by an experienced microscopist following a routine procedure of 10% Giemsa stained thick and thin blood film under oil immersion examination of parasites at 100X magnification at TGH. The thick film was used for parasite quantification as the thin blood film was used for species identification.

### Ethical csonsiderations

The Institutional Review Board of University of Cape Coast granted approval (UCCIRB/CHAS/2022/146) for the study. In addition, a written informed consent was obtained from each study participant. Above all, the study protocols were in strict adherence to the ethical standards of the TGH, Ghana Health Service and the World Medical Association Declaration of Helsinki.

### Data analysis

Statistical Package for the Social Sciences (SPSS IBM, USA) version 25 was used for the data analysis. Data from continuous variables were log-transformed to improve normality if the distribution deviated from normality before analysis and presented as mean ± standard deviation. The normal Q-Q plot was used in assessing normality of data distribution. However, categorical variables were presented in percentages and compared by chi-square test of independence. Each study group was divided into two based on malarial status and mean levels of continuous variable were compared across groups by one-way analysis of variance (ANOVA) followed by the least significant difference (LSD) *post-hoc* test. The Cohen's d, ω and omega-squared with fixed effect were estimated as the effect sizes for t-test, chi-squared test and ANOVA comparison of mean values respectively. Their respective standard thresholds were applied in interpreting the results. Bivariate correlation test was applied to assess the extent of linear relationship among measured indices. A binary logistic regression analysis that controlled for relevant confounders was applied to identify predictors of insulin resistance. In all analyses, statistical significance was determined at $P < 0.05$.

## Results

Participants with T2DM had lived with the condition for between 2 and 25 years with an average duration of 8.5 years and had been attending the diabetes clinic of TGH for care. However, participants without diabetes were at TGH for different health challenges or accompanying a sick family member for medical care.

Table 1 shows the sociodemographic and clinical characteristics of study participants including age, gender, anthropometry, and exercise and medication intake. The distribution of diabetic and non-diabetic respondents into categories based on hip circumference, waist-to-hip ratio and the intake of cholesterol medication did not differ significantly ($P > 0.05$) in proportion, but the differences in distribution were significant ($P < 0.05$) across the categorizations of the other variables (Table 1). Generally, the strengths of the associations ranged from medium (Cohen's $\omega \geq 0.3$) to high (Cohen's $\omega \geq 0.5$) for parameters with statistically significant difference ($P < 0.05$) in proportions of participants across categories. However,

**Table 1. Socio-demographic and clinical characteristics of participants.**

| Parameters | Patients With Diabetes | Patients Without Diabetes | P Value | Effect size |
|---|---|---|---|---|
| Age (years) | N (%) | N (%) | | 0.531 |
| 20–29 | 3 (3.8) | 17 (21.3) | < 0.001* | |
| 30–39 | 7 (8.8) | 15 (18.8) | | |
| 40–49 | 18 (22.5) | 30 (37.5) | | |
| 50–59 | 23 (28.7) | 15 (18.8) | | |
| 60–69 | 25 (31.3) | 3 (3.8) | | |
| ≥ 70 | 4 (5) | 0 (0) | | |
| Gender | N (%) | N (%) | | 0.382 |
| Male | 31 (38.8) | 50 (62.5) | 0.003* | |
| Female | 49 (61.3) | 30 (37.5) | | |
| Body Mass Index | N (%) | N (%) | | 0.356 |
| Normal | 15 (18.8) | 28 (35) | 0.012* | |
| Overweight | 34 (42.5) | 36 (45) | | |
| Obese | 31 (38.8) | 16 (20) | | |
| Systolic Blood Pressure | N (%) | N (%) | | 0.563 |
| Normotension | 6 (7.5) | 27 (33.8) | < 0.001* | |
| Pre-hypertension | 32 (40.0) | 38 (47.5) | | |
| Hypertension | 42 (52.5) | 15 (26.3) | | |
| Diastolic blood pressure | N (%) | N (%) | | 0.495 |
| Normotension | 30 (37.5) | 54 (67.5) | < 0.001* | |
| Pre-hypertension | 24 (30.0) | 16 (20.0) | | |
| Hypertension | 26 (32.5) | 10 (12.5) | | |
| Waist circumference | N (%) | N (%) | | 0.421 |
| Normal | 40 (50) | 60 (75) | 0.001* | |
| High | 40 (50) | 20 (25) | | |
| Hip circumference | N (%) | N (%) | | 0.01 |
| Low | 3 (3.8) | 5 (6.3) | 0.721 | |
| Normal | 72 (90.0) | 69 (86.3) | | |
| High | 5 (6.3) | 6 (7.5) | | |
| Waist-to-hip ratio | N (%) | N (%) | | 0.018 |
| Normal | 49 (61.3) | 58 (72.5) | 0.131 | |
| High | 31 (38.8) | 22 (27.5) | | |
| Takes Hypertension Medication | N (%) | N (%) | | 0.412 |
| Yes | 64 (80) | 20 (25) | < 0.001* | |
| No | 16 (20) | 60 (75) | | |
| Takes Cholesterol Medication | N (%) | N (%) | | 0.092 |
| Yes | 2 (2.5) | 7 (8.8) | 0.087 | |
| No | 78 (97.5) | 73 (91.2) | | |
| Regular exercise | N (%) | N (%) | | 0.425 |
| Less often | 55 (68.8) | 17 (21.3) | < 0.001* | |
| Often | 25 (31.3) | 62 (77.5) | | |
| Very often | 0 (0.0) | 1 (1.3) | | |

parameters without any difference (P>0.05) in proportion of participants across categories demonstrated a relatively small strength of association (Cohen's ω≤0.1; Table 1). Participants with T2DM were predominantly aged 60–69 years, female, overweight/obese, taking hypertensive medication and exercised less often in comparison with their non-diabetic counterparts who were predominantly aged 40–49 years, male, and exercised more often.

In table 2, the participants were divided into four groups for comparison of levels of measured indices. Apart from the anthropometric indices, haemoglobin, RBC, MCH, MCHC and monocyte levels that did not differ across groups, all the other indices differed significantly across the four groups of respondents (P<0.05; Table 2). Parasite level was higher (P<0.001; Table 2) in patients without diabetes with a small strength of the association (Cohen's d=0.334; Table 2) compared with their diabetic counterparts. In general, the strengths of the associations were quite high for age, systolic blood pressure, lipid profile, galectin-3, eosinophil, platelet, insulin, HOMA-IR and HOMA-B ($\omega^2 \geq 0.14$; Table 2). The strength was medium for the rest of the parameters that demonstrated a difference in mean values for the ANOVA test ($0.06 \leq \omega^2 < 0.14$; Table 2). Expectedly, parameters that did not differ in mean levels showed a small strength of association ($\omega^2 < 0.01$; Table 2). However, the post-hoc tests revealed participants with T2DM to be older with higher systolic blood pressure (P<0.05; Table 2). Levels of total cholesterol, LDL and HDL were higher but triglyceride level was lower in the participants with malaria irrespective of diabetes status (all P<0.05; Table 2). Interestingly, malaria patients without diabetes exhibited lower (all P<0.001; Table 2) platelet, lymphocyte and eosinophil counts. Serum galectin-3 level was generally higher in participants with malaria and lowest in non-diabetic participants without malaria. The observed levels of insulin, insulin resistance and beta-cell function were highest for participants with T2DM without malaria (Table 2).

In order to explore linear relationships among measured parameters, a bivariate correlation was conducted in each group after a partial correlation in the entire sample that controlled for diabetes and malaria status. As expected, the observed correlations among the measured parameters differed with group and the entire sample. For instance, age correlated (P<0.05; Table 3A) positively with galectin-3 and triglyceride but negatively with HDL in the entire sample irrespective of malaria or diabetes status yet no correlation (P>0.05; Table 3B) was observed between age and any of the measured parameters in the diabetic patients without malaria. Additionally, galectin-3 correlated negatively (r=−0.391; P<0.05; Table 3B) with HOMA-B in diabetic participants with malaria but positively with age (r=0.35; P<0.05; Table 3C) in diabetic patients without malaria and HOMA-B (r=0.321; P<0.05) and insulin (r=0.322; P<0.05) in malaria patients without diabetes (Table 3D). Interestingly, the positive trend of correlation of insulin with HOMA-IR and HOMA-B was observed across the study groups as in the entire sample ($r_{range}$=0.612–0.985; P<0.05; Table 3). A positive correlation (r=0.316; P<0.05; Table 3D) between BMI and galectin-3 was observed only in the non-diabetic patients with malaria.

To further explore the association between insulin resistance defined as HOMA-IR>2.6, and the other measured parameters, binary logistic regression analyses were conducted for the entire participants irrespective of their health status. The results showed that insulin resistance could associate with the circulating levels of glucose (*B*=0.603, *Wald*=10.52, *Exp (B)* = 1.83, CI: 1.27–2.63; P=0.001; Table 4) and insulin (*B*=1.145, *Wald*=18.61, *Exp (B)* = 3.14, CI: 1.87–5.23; P<0.001; Table 4) only. For a unit rise in glucose, the odds of having insulin resistance increased by 83% in our participants. In terms of insulin, a unit increase demonstrated a 214% increase in the odds of having insulin resistance. The model could explain 67.7% (Cox & Snell $R^2$=0.677) to 91% (Nagelkerke $R^2$=0.91) of the observed variation in insulin resistance in our sample. Omnibus test of the model was statistically significant ($X^2$=180.604, df=5; P<0.001) but the Hosmer and Lemeshow test was not significant ($X^2$=6.657, df=8; P=0.574), suggesting that our model fit was appropriate.

## Discussion

In this study, the relationship between galectin-3 and insulin resistance was investigated in diabetic and non-diabetic participants with or without malaria receiving treatment at the TGH. Generally, our results demonstrated that participants with diabetes were older with higher levels of blood pressure and insulin resistance than their non-diabetic counterparts.

**Table 2. Comparison of Mean Levels of Measured Parameters by Group.**

| PARAMETER | NDM | DMM | DMWM | NDWM | P-value | ES |
|---|---|---|---|---|---|---|
| Age (years) | 37.45±11.85 | 53.13±12.44 | 53.95±11.86 | 44.15±9.37 | <0.001* | 0.251 |
| Height (m) | 1.7±0.06 | 1.68±0.04 | 1.69±0.04 | 1.68±0.05 | 0.064 | 0.027 |
| Weight (Kg) | 78.96±17.31 | 81.54±16.46 | 85.47±20.29 | 77.43±13.74 | 0.171 | 0.013 |
| Waist (cm) | 35.03±5.71 | 36.00±4.94 | 36.25±4.13 | 34.13±4.08 | 0.176 | 0.012 |
| Hip (cm) | 40.63±5.97 | 42.33±5.00 | 42.20±4.93 | 40.38±4.47 | 0.192 | 0.011 |
| BMI (kg/m²) | 26.75±6.00 | 28.38±5.39 | 29.65±6.96 | 27.00±5.62 | 0.118 | 0.018 |
| WHR | 0.87±0.09 | 0.85±0.06 | 0.87±0.07 | 0.85±0.08 | 0.408 | −0.001 |
| SBP (mmHg) | 127.63±17.00 | 141.88±18.55 | 145.35±17.32 | 127.90±18.49 | <0.001* | 0.155 |
| DBP (mmHg) | 75.08±15.38 | 86.85±14.05 | 82.23±11.38 | 75.73±11.70 | <0.001* | 0.104 |
| Parasite/μL | 9981±4.00 | 3819±5.00 | – | – | <0.001*γ | 0.334 |
| WBC x (10⁹/l) | 4.99±1.55 | 5.66±1.81 | 6.52±2.19 | 5.67±1.97 | 0.006* | 0.059 |
| RBC x (10¹²/l) | 4.79±0.60 | 4.49±0.65 | 4.76±0.61 | 4.76±0.54 | 0.092 | 0.022 |
| HB (mg/L) | 12.75±1.74 | 12.13±1.99 | 12.85±1.56 | 12.65±1.63 | 0.249 | 0.007 |
| HCT (%) | 38.22 ±4.75 | 37.24 ±5.45 | 39.48 ±4.10 | 37.89 ±4.07 | 0.179 | 0.012 |
| MCV (fl) | 80.01 ±4.79 | 83.09 ±4.62 | 82.96 ±5.89 | 80.09 ±6.01 | 0.008* | 0.055 |
| MCH (pg/cell) | 32.40 ±2.10 | 32.30 ±1.45 | 28.31±2.42 | 27.55 ±2.66 | 0.146 | 0.015 |
| MCHC (g/dl) | 34.41 ±1.38 | 34.23 ±1.25 | 33.75 ±1.37 | 34.25 ±1.76 | 0.21 | 0.010 |
| Platelet x (10⁹/l) | 116.78±52.77 | 139.78±49.21 | 254.58±46.94 | 241.03±54.28 | <0.001* | 0.583 |
| Neutrophil | 3.34±1.46 | 3.97±1.38 | 3.27±1.87 | 2.82±1.78 | 0.02* | 0.043 |
| Lymphocyte | 1.06±0.66 | 1.28±0.99 | 2.49±0.87 | 2.15±0.70 | <0.001* | 0.337 |
| Monocyte | 0.52±0.30 | 0.45±0.32 | 0.57±0.25 | 0.50±0.21 | 0.255 | 0.007 |
| Eosinophil | .04±0.07 | .05±0.08 | 0.15±0.13 | 0.19±0.17 | <0.001* | 0.213 |
| Basophil | 0.03±0.06 | .03±0.03 | 0.06±0.11 | 0.08±0.12 | 0.039* | 0.034 |
| Cholesterol (mmol/l) | 3.99±0.91 | 4.52±1.10 | 5.29±1.36 | 5.53±1.17 | <0.001* | 0.213 |
| LDL (mmol/l) | 2.45±0.76 | 2.67±1.07 | 3.32±1.19 | 3.53±0.97 | <0.001* | 0.15 |
| HDL (mmol/l) | 0.62±0.38 | 0.87±0.50 | 1.35±0.58 | 1.40±0.59 | <0.001* | 0.275 |
| Triglyceride (mmol/l) | 2.00±1.82 | 2.03±0.93 | 1.35±0.71 | 1.02±0.54 | <0.001* | 0.117 |
| FPG (mmol/l) | 5.66±1.01 | 12.59±3.47 | 9.96±3.94 | 5.93±1.22 | <0.001* | 0.523 |
| Galectin-3 (ng/mL) | 12.65±5.21 | 13.76±5.12 | 10.36±4.77 | 7.72±3.58 | <0.001* | 0.182 |
| Insulin (mIU/l) | 8.34±3.36 | 5.95±2.19 | 26.20±2.75 | 7.76±2.42 | <0.001* | 0.294 |
| HOMA-IR | 4.62±6.17 | 4.27±3.71 | 17.77±15.11 | 3.14±4.04 | <0.001* | 0.317 |
| HOMA-B (%) | 85.92±2.92 | 15.11±2.70 | 96.01±2.71 | 68.42±3.40 | <0.001* | 0.177 |

**Statistically significant *post-hoc* tests results**

| VARIABLE | NDM vs DMM | NDM vs DMWM | NDM vs NDWM | DMM vs DMWM | DMM vs NDWM | DMWM vs NDWM |
|---|---|---|---|---|---|---|
| Age | <0.001 | <0.001 | – | – | 0.004 | 0.001 |
| SBP | 0.003 | <0.001 | – | – | 0.004 | <0.001 |
| DBP | <0.001 | – | – | – | 0.001 | – |
| WBC | – | 0.003 | – | – | – | – |
| Platelet | – | <0.001 | <0.001 | <0.001 | <0.001 | – |
| MCV | 0.011 | 0.015 | – | – | 0.014 | 0.018 |
| Neutrophil | – | – | – | – | 0.011 | – |
| Lymphocyte | – | <0.001 | <0.001 | <0.001 | <0.001 | – |
| Eosinophil | – | <0.001 | <0.001 | <0.001 | <0.001 | – |
| Basophil | – | – | 0.023 | – | 0.013 | – |
| Cholesterol | – | <0.001 | <0.001 | 0.017 | <0.001 | – |

*(Continued)*

Table 2. (Continued)

| PARAMETER | NDM | DMM | DMWM | NDWM | P-value | ES |
|---|---|---|---|---|---|---|
| LDL | – | <0.001 | <0.001 | 0.029 | 0.001 | – |
| HDL | – | <0.001 | <0.001 | <0.001 | <0.001 | |
| Triglyceride | – | – | <0.001 | 0.038 | <0.001 | – |
| FPG | <0.001 | <0.001 | – | <0.001 | <0.001 | <0.001 |
| Galectin-3 | – | 0.031 | <0.001 | 0.002 | <0.001 | 0.013 |
| Insulin | 0.032 | <0.001 | – | <0.001 | – | <0.001 |
| HOMA-IR | – | <0.001 | – | <0.001 | – | <0.001 |
| HOMA-B | <0.001 | – | 0.017 | <0.001 | 0.004 | 0.028 |

*FPG: fasting plasma glucose; NDM: Non-diabetic with malaria; DMM: Diabetes mellitus with malaria; DMWM: Diabetes mellitus without malaria; NDWM: Non-diabetes without malaria; Least significant difference (LSD); \*: significant p-value by analysis of variance (ANOVA); LDL: low-density lipoprotein cholesterol; HDL: high-density lipoprotein cholesterol; LDL: Low-density lipoprotein cholesterol; DBP: diastolic blood pressure; SBP: systolic blood pressure; FPG: fasting plasma glucose; BMI: body mass index; TyG: triglyceride-glucose index; HOMA-IR: homeostatic model assessment of insulin resistance; HOMA-B: homeostatic model assessment of beta-cell function; \*γ: significant p-value by t-test; WBC: white blood cell; RBC: red blood cell; MCV: mean corpuscular volume; MCHC: mean corpuscular haemglobin concentration; MCH: mean corpuscular haemoglobin; HCT: haematocrit; HB: haemoglobin; ES: effect size determined as omega-squared fixed effect except the parasite level which was determined as Cohan's d.*

However, *Plasmodium falciparum* parasite level (9981 versus 3891) was higher in the nondiabetic group compared to their diabetic counterpart. Insulin resistance associated with circulating insulin and glucose levels in the entire sample irrespective of diabetes or malaria status. Galectin-3 could not associate with insulin resistance in our context.

The finding of diabetic participants being older is in corroboration with previous studies [22–24] that involved larger sample sizes. Indeed, T2DM predominantly occurs after 40 years and the diabetic condition is underpinned by insulin resistance which also promotes increased blood pressure which are all components of metabolic syndrome [25]. In a recent review, Jia and Sowers [26] identified undue renin-angiotensin-aldosterone, sympathetic nervous systems activation, oxidative stress, inflammation and mitochondrial dysfunction as factors that link hyperglycemia to increased blood pressure. Additionally, increased epithelial sodium channels, altered gut microbiota, extracellular vesicles and raised sodium-glucose cotransporter activity were implicated as part of the molecular mechanisms explaining the relationship between hyperglycemia and increased blood pressure [26].

Although the risk of insulin resistance is known to increase with age [27], in the current study, a negative correlation was rather observed between age and HOMA-IR, WHR, HDL and insulin but positive with galectin-3 in respondents with T2DM without malaria. The negative correlation of age with HOMA-IR in our diabetic respondents without malaria corroborates a previous study [28] that involved 169 non-diabetic Chinese respondents and could be ascribed to reduced insulin secretion and physical activity with age. As one ages, the risk of beta-cell stress and apoptosis increases resulting in reduced glucose-stimulated insulin release [29]. Although pharmacological approach to increasing HDL concentration appears more attractive, there is some consensus that physical activity improves HDL concentration and functionality [30,31]. This implies that reduced physical activity which normally occurs with age could result in impaired HDL concentration as observed in the diabetic respondents without malaria in the current study. Interestingly, the diabetic group without malaria exhibited the highest levels of insulin and insulin resistance, which can be adduced to a probable reduced rate of insulin clearance [28], coupled with unimpaired beta-cell secretory function in the mist of impaired insulin-mediated signaling for glucose uptake. With time, the secretory function of the beta cells could reduce drastically due to probable cell exhaustion and glucotoxicity [4,32]. This situation may aggravate the hyperglycaemic situation and related complications if appropriate treatment measures are not taken in due course.

**Table 3. Statistically significant correlations among measured indices by study group.**

| Parameter | Age | HOMA-IR | HOMA-B | Insulin | Galectin-3 | WHR | BMI | Chol | LDL | HDL | Trig |
|---|---|---|---|---|---|---|---|---|---|---|---|
| **A. Entire Participants irrespective of diabetes or malaria status** | | | | | | | | | | | |
| Age | 1 | – | – | – | 0.329 | – | – | – | – | −0.196 | 0.208 |
| HOMA-IR | | 1 | 0.412 | 0.902 | – | 0.179 | 0.334 | 0.241 | 0.186 | 0.169 | – |
| HOMA-B | | | 1 | 0.612 | – | – | – | – | – | – | – |
| Insulin | | | | 1 | – | – | 0.314 | 0.258 | 0.27 | – | – |
| Galectin-3 | | | | | 1 | – | – | – | – | – | – |
| WHR | | | | | | 1 | 0.28 | 0.184 | – | – | 0.194 |
| BMI | | | | | | | 1 | 0.254 | 0.235 | – | 0.198 |
| Cholesterol | | | | | | | | 1 | 0.848 | 0.396 | 0.191 |
| LDL | | | | | | | | | 1 | 0.265 | −0.24 |
| HDL | | | | | | | | | | 1 | −0.216 |
| Trig | | | | | | | | | | | 1 |
| **B. Patients with diabetes and malaria** | | | | | | | | | | | |
| Parameter | Age | HOMA-IR | HOMA-B | Insulin | Galectin-3 | WHR | BMI | Chol | LDL | HDL | Trig |
| Age | 1 | – | – | – | – | – | – | – | – | – | – |
| HOMA-IR | | 1 | 0.543 | 0.875 | – | – | – | – | – | – | – |
| HOMA-B | | | 1 | 0.861 | −0.391 | – | – | – | – | – | – |
| Insulin | | | | 1 | – | – | – | – | – | – | – |
| Galectin-3 | | | | | 1 | – | – | – | – | – | – |
| WHR | | | | | | 1 | 0.516 | – | −0.32 | – | – |
| BMI | | | | | | | 1 | – | – | – | – |
| Cholesterol | | | | | | | | 1 | 0.857 | 0.418 | – |
| LDL | | | | | | | | | 1 | – | −0.395 |
| HDL | | | | | | | | | | 1 | – |
| Trig | | | | | | | | | | | 1 |
| **C. Patients with diabetes without malaria** | | | | | | | | | | | |
| Parameter | Age | HOMA-IR | HOMA-B | Insulin | Galectin-3 | WHR | BMI | Chol | LDL | HDL | Trig |
| Age | 1 | −0.579 | – | −0.356 | 0.35 | −0.315 | – | – | – | −0.401 | – |
| HOMA-IR | | 1 | – | 0.882 | – | 0.314 | 0.498 | 0.386 | – | – | – |
| HOMA-B | | | 1 | 0.985 | – | – | – | – | – | – | – |
| Insulin | | | | 1 | – | – | 0.564 | 0.43 | 0.457 | – | – |
| Galectin-3 | | | | | 1 | – | – | – | – | – | – |
| WHR | | | | | | 1 | – | – | – | – | – |
| BMI | | | | | | | 1 | – | 0.415 | – | – |
| Cholesterol | | | | | | | | 1 | 0.878 | 0.436 | 0.453 |
| LDL | | | | | | | | | 1 | – | – |
| HDL | | | | | | | | | | 1 | 0.258 |
| Trig | | | | | | | | | | | 1 |
| **D. Non-diabetic patients with malaria** | | | | | | | | | | | |
| Parameter | Age | HOMA-IR | HOMA-B | Insulin | Galectin-3 | WHR | BMI | Chol | LDL | HDL | Trig |
| Age | – | – | – | – | – | – | – | – | – | −0.36 | – |
| HOMA-IR | | 1 | 0.844 | 0.973 | – | – | – | – | – | – | – |
| HOMA-B | | | 1 | 0.923 | 0.321 | – | – | – | – | – | – |
| Insulin | | | | 1 | 0.322 | – | – | – | – | – | – |
| Galectin-3 | | | | | 1 | – | 0.316 | – | – | – | – |
| WHR | | | | | | 1 | – | – | – | – | – |

*(Continued)*

**Table 3.** (Continued)

| Parameter | Age | HOMA-IR | HOMA-B | Insulin | Galectin-3 | WHR | BMI | Chol | LDL | HDL | Trig |
|---|---|---|---|---|---|---|---|---|---|---|---|
| BMI | | | | | | | 1 | – | – | – | – |
| Cholesterol | | | | | | | | 1 | 0.797 | 0.359 | – |
| LDL | | | | | | | | | 1 | 0.483 | −0.327 |
| HDL | | | | | | | | | | 1 | −0.565 |
| Trig | | | | | | | | | | | 1 |
| **E.Non-diabetic patients without malaria** | | | | | | | | | | | |
| Parameter | Age | HOMA-IR | HOMA-B | Insulin | Galectin-3 | WHR | BMI | Chol | LDL | HDL | Trig |
| Age | – | – | – | – | – | – | – | – | – | – | 0.489 |
| HOMA-IR | | 1 | 0.396 | 0.951 | – | 0.334 | – | – | – | – | 0.492 |
| HOMA-B | | | 1 | 0.618 | – | – | – | – | – | – | – |
| Insulin | | | | 1 | – | – | – | – | – | – | 0.493 |
| Galectin-3 | | | | | 1 | – | – | – | – | – | – |
| WHR | | | | | | 1 | – | 0.412 | 0.423 | – | – |
| BMI | | | | | | | 1 | 0.417 | 0.336 | – | – |
| Cholesterol | | | | | | | | 1 | 0.851 | 0.472 | – |
| LDL | | | | | | | | | 1 | – | – |
| HDL | | | | | | | | | | 1 | – |
| Trig | | | | | | | | | | | 1 |

*Trig: triglyceride; LDL: low-density lipoprotein cholesterol; HDL: high-density lipoprotein cholesterol; LDL: Low-density lipoprotein cholesterol; WHR: waist-to-hip ratio; FPG: fasting plasma glucose; BMI: body mass index; HOMA-IR: homeostatic model assessment of insulin resistance; HOMA-B: homeostatic model assessment of beta-cell function*

**Table 4. Predictors of insulin resistance in the participants.**

| Variable | B | SE | Wald | Exp(B) | 95% CI for Exp(B) | P-value |
|---|---|---|---|---|---|---|
| Constant | −16.576 | 3.84 | 18.60 | 0.00 | | <0.001* |
| FPG | 0.603 | 0.19 | 10.52 | 1.83 | 1.27–2.63 | 0.001* |
| Galectin-3 | 0.077 | 0.08 | 0.83 | 1.08 | 0.92–1.27 | 0.362 |
| Age | 0.031 | 0.04 | 0.60 | 1.03 | 0.95–1.12 | 0.44 |
| Insulin | 1.145 | 0.27 | 18.61 | 3.14 | 1.87–5.23 | <0.001* |
| HOMA-B | −0.002 | 0.01 | 0.04 | 1.00 | 0.98–1.02 | 0.851 |

*FPG: fasting plasma glucose; HOMA-B: homeostatic model assessment of beta-cell function; B: regression coefficient; SE: standard error; Exp: exponent; CI: confidence interval;*

*: significant p-value.*

Diabetic respondents with malaria demonstrated a higher plasma glucose level than their counterparts without malaria, yet in the non-diabetic groups, circulating glucose levels were comparable. This observation, coupled with the lack of an association between circulating glucose and parasite levels suggests that reduced beta-cell secretory function and impaired insulin function rather than malaria are responsible for this. In a longitudinal Ghanaian study, Acquah et al. [3] demonstrated malaria-induced increased plasma glucose level only in semi-immune adult non-diabetic respondents as opposed to their diabetic counterparts. Interestingly, this finding could not be corroborated in the non-diabetic respondents of the current study due probably to differences in study design. This notwithstanding, a careful examination of the beta-cell function appears to point to an increased risk of development of T2DM in the non-diabetic respondents without

malaria through impaired secretory function of their beta cells. However, a longitudinal study involving larger sample may be appropriate in determining the actual risk in this group to guide appropriate policy response. Moreover, the contribution of HOMA-B in partly accounting for the observed insulin resistance, depicted by HOMA-IR was clearly demonstrated in all study groups except the diabetic respondents who did not get malaria in the current study. This calls for a probable varied treatment strategies for the two groups of diabetic respondents for the effective management of the condition.

A number of earlier studies in humans [12,14,33] have reported galectin-3 level to be higher in diabetics and found to impair insulin signaling. In a large sample epidemiological study involving 6,586 respondents of different racial backgrounds in the United State of America, Vora et al. [34] demonstrated positive associations of galectin-3 with prevalence and incidence of diabetes. In a mice study, Oakley et al. [16] observed galectin-3 to facilitate cerebral malaria. In the current study, circulating galectin-3 level tended to be higher in diabetics compared with non-diabetic controls without malaria in support of previous studies [12,14,33] but at variance with a Japanese study that rather associated low galectin-3 level with insulin resistance [13]. The discordant observation between the current study and the previous one [13] could be due to differences in sample size as the previous study included only 20 T2DM patients. Irrespective of malaria or diabetes status, galectin-3 correlated positively with age in the entire sample as it did in diabetic respondents who did not get malaria. This positive correlational trend was observed between galectin-3 and insulin as well as HOMA-B in the non-diabetic respondents with malaria. In diabetic patients who had malaria, a negative correlation was found between galectin-3 and HOMA-B. In as much as these results give further credence to the probable diabetogenic role of galectin-3 in our sample, irrespective of disease background, they point to different mechanisms by which malaria associates with the different groups of respondents. Thus, it appears that in malaria, increased galectin-3 increased HOMA-B in the non-diabetic respondents but reduced it in the diabetic respondents. Therefore, we speculate that in malaria, galactin-3 may associate with worsening hyperglycaemia through impaired insulin release in diabetics but heightens beta-cell secretory function in non-diabetic respondents. In an American study that involved 486 people of African descent, Ishimwe et al. [35] observed beta-cell failure (62%) to be the predominant cause of abnormal glucose tolerance instead of insulin resistance (38%). In a number of recent reviews [36,37], three mechanisms of beta-cell failure have been proposed which include reduced beta-cell number through cell death, stress-induced cell exhaustion and identity loss through loss of gene expression. With malaria being implicated in a number of these mechanisms [3,5,38] coupled with the possibility of multiple bouts of malaria in a lifetime, the seeming increased HOMA-B with increased galectin-3 observed in the non-diabetic respondents who had malaria in the current study connotes an increased diabetogenic risk. However, in our context, results from the regression analyses did not identify galectin-3 as an independent predictor of insulin resistance. This implies that galectin-3 is not linearly associated with insulin resistance in our context. This lack of association between galectin-3 and insulin resistance in the current study appears to be at variance with previous studies [12–14,33,34] that associated galectin-3 with insulin signaling and incidence and prevalence of diabetes. Differences in study design, sample size and characteristics of study participants between the current study and the previous ones [12–14,33,34] may account for the differences in findings.

Interestingly, a number of studies [39–41] have associated blood cell parameters with diabetes risk and complications in different populations. Indeed diabetes is thought to weaken the immune response capability of the affected making them more susceptible to infection [42]. We observed a lower parasite density in patients with diabetes compared with their non-diabetic counterparts contrary to previous studies [43,44] that reported higher parasitaemia in diabetes patients. Our findings support those of previous studies [45,46] that found lower parasite levels in patients with diabetes. The reduced parasite level in diabetes patients with malaria could be due to cytotoxic effects of glucose-induced peroxides on the *Plasmodium* parasites [47–50] which is heightened in diabetic compared with the non-diabetic state. However, the higher parasite levels in patients with diabetes in the earlier reports [43,44] may be attributed to higher glucose levels that promote parasite growth since the *Plasmodium* parasite lacks the requisite enzymes for gluconeogenesis [51]. Interestingly, diabetes is known to impair immune function [42,52] making patients with diabetes more susceptible to *Plasmodium* infection [43] and disease severity [44]. To this end, our finding supports the view of weakened immunity of

diabetic respondents [42,52] in that they could not tolerate high parasite numbers to develop clinical signs and symptoms of malaria in spite of comparable levels of components of the white blood cells with their non-diabetic counterparts. The non-diabetic group, being immunologically stronger, needed high parasite numbers to demonstrate detectable signs and symptoms of clinical significance to warrant appropriate treatment. These results suggest that the diabetic and non-diabetic participants responded differently to the *Plasmodium falciparum* parasite infection in the current study. Nonetheless, the observed positive association of insulin resistance with circulating glucose and insulin levels in our context reinforces the heightened diabetogenic risk for the non-diabetic participants and risk of probable complication for the diabetic group if the levels of these indices are not properly controlled.

The study is limited by its cross-sectional design that makes establishment of causality impossible. Secondly, the relatively small sample size makes undue generalization of findings challenging. Moreover, the possibility of bias in sample selection in spite of the randomized selection of participants cannot be completely ruled out. Above all, the use of surrogate markers that rely on fasting parameters makes assessment of postprandial insulin sensitivity difficult due to fluctuation of fasting insulin and glucose levels. The use of glucose tolerance test may have provided a better view of the body's response to postprandial insulin in malaria. In spite of the above limitations, our findings have demonstrated a probable indirect role for galectin-3 in malaria-related insulin resistance in diabetic and non-diabetic participants in the Ghanaian setting.

## Conclusion

Although galectin-3 levels increased in malaria patients irrespective of diabetes status, its relationship with insulin resistance appears more complex than a linear fashion. Irrespective of malaria or T2DM status, glucose and insulin levels associated positively with insulin resistance.

Further studies are needed to properly define the exact role of galectin-3 in the evolution of T2DM in malaria-endemic regions of the globe.

## Acknowledgments

We express our profound gratitude to the management and staff of Tema General Hospital for their support in allowing their facility to be used for the successful conduct of the study.

## Author contributions

**Conceptualization:** Emmanuel Nortey, Leonard Derkyi-Kwarteng, Daniel Amoako-Sakyi, Ansumana Sandy Bockarie, Samuel Acquah.

**Data curation:** Emmanuel Nortey, Leonard Derkyi-Kwarteng, Daniel Amoako-Sakyi, Ansumana Sandy Bockarie, Samuel Yeboah, Samuel Acquah.

**Formal analysis:** Emmanuel Nortey, Leonard Derkyi-Kwarteng, Daniel Amoako-Sakyi, Ansumana Sandy Bockarie, Samuel Acquah.

**Funding acquisition:** Emmanuel Nortey.

**Investigation:** Emmanuel Nortey, Leonard Derkyi-Kwarteng, Ansumana Sandy Bockarie, Samuel Acquah.

**Methodology:** Emmanuel Nortey, Leonard Derkyi-Kwarteng, Daniel Amoako-Sakyi, Ansumana Sandy Bockarie, Samuel Yeboah, Samuel Acquah.

**Project administration:** Emmanuel Nortey, Leonard Derkyi-Kwarteng, Daniel Amoako-Sakyi, Samuel Acquah.

**Resources:** Emmanuel Nortey, Samuel Yeboah, Samuel Acquah.

**Software:** Samuel Yeboah.

**Supervision:** Leonard Derkyi-Kwarteng, Daniel Amoako-Sakyi, Ansumana Sandy Bockarie, Samuel Acquah.

**Validation:** Emmanuel Nortey, Ansumana Sandy Bockarie, Samuel Yeboah, Samuel Acquah.

**Visualization:** Samuel Yeboah, Samuel Acquah.

**Writing – original draft:** Leonard Derkyi-Kwarteng, Ansumana Sandy Bockarie, Samuel Acquah.

**Writing – review & editing:** Leonard Derkyi-Kwarteng, Daniel Amoako-Sakyi, Ansumana Sandy Bockarie, Samuel Yeboah, Samuel Acquah.

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
