## [Decision Letter · Decision Letter 0]

27 May 2025

PONE-D-25-15329Galectin-3 did not associate with malaria-related insulin resistance in diabetic and non-diabetic respondents at a Ghanaian General HospitalPLOS ONE

Dear Dr. Acquah,

Thank you for submitting your manuscript to PLOS ONE. After careful consideration, we feel that it has merit but does not fully meet PLOS ONE’s publication criteria as it currently stands. Therefore, we invite you to submit a revised version of the manuscript that addresses the points raised during the review process.

A rebuttal letter that responds to each point raised by the academic editor and reviewer(s). You should upload this letter as a separate file labeled 'Response to Reviewers'.A marked-up copy of your manuscript that highlights in YELLOW changes made to the original version. You should upload this as a separate file labeled 'Revised Manuscript with Track Changes'.An unmarked version of your revised paper without tracked changes. You should upload this as a separate file labeled 'Manuscript'.

We look forward to receiving your revised manuscript.

Kind regards,

Mohammad Reza Mahmoodi, Ph.D.

Academic Editor

PLOS ONE

Reviewers' comments:

Reviewer's Responses to Questions

**Comments to the Author**

1. Is the manuscript technically sound, and do the data support the conclusions?

Reviewer #1: Partly

Reviewer #2: Partly

2. Has the statistical analysis been performed appropriately and rigorously? 

Reviewer #1: Yes

Reviewer #2: Yes

3. Have the authors made all data underlying the findings in their manuscript fully available?

Reviewer #1: Yes

Reviewer #2: Yes

4. Is the manuscript presented in an intelligible fashion and written in standard English?

Reviewer #1: No

Reviewer #2: Yes

5. Review Comments to the Author

Reviewer #1: Dear Authors,

Your manuscript addresses an important topic and contains valuable data. However, before it can be considered for publication, it needs to address several major concerns related to structure, clarity, and scientific rigor. Below are my detailed comments and questions:

1. Line Numbers: The manuscript lacks line numbers, which makes it difficult to refer to specific sections. Please include them in the revised version to facilitate the review process.

2. Language and Clarity: The manuscript requires extensive language editing. Many sentences are overly long, ambiguous, or grammatically incorrect. We strongly recommend having a native English speaker or a professional academic editor review and edit the manuscript.

3. The abstract should be rewritten for clarity and completeness.

4. Phrases like “could neither predict” are vague and need rephrasing for clarity.

5. Please clearly state whether Galectin-3 elevation is indicative of an inflammatory state or if it reflects an independent mechanism.

6. The statement, “Galectin-3 is a chimeric galectin found in vertebrates,” is unclear and should be rewritten for accuracy and clarity.

7. Clearly specify the type of study conducted (e.g., cross-sectional, observational) early in the introduction.

8. It would strengthen the manuscript to include data on the prevalence and burden of diabetes and malaria in your specific country or region, if available.

9. The hypothesis at the end of the introduction is vague and poorly articulated. A more focused and testable hypothesis is needed, as the current ambiguity negatively affects the coherence of the entire manuscript.

10. The inclusion and exclusion criteria require clearer explanation. For example:

• Why were people who had a COVID-19 vaccination history and those who had received vaccinations excluded?

• What is the regional COVID-19 vaccination policy, and are there vaccines in use that could mimic natural infections?

11. It is unclear whether participants had active malaria infections at the time of enrollment or only past exposure. Please clarify.

12. You should explicitly state the rationale for excluding individuals with autoimmune diseases such as rheumatoid arthritis (RA) and type 1 diabetes mellitus (T1DM).

13. You mention calculating prevalence ratios for malaria and diabetes. Was it assumed these groups were independent? If comparisons were made between diabetics and non-diabetics with and without malaria, the difference needs to be explicitly and clearly described.

14. The threshold used to define insulin resistance (HOMA-IR > 2.6) should be referenced and justified, ideally based on prior studies in a comparable population.

15. No details were provided about whether important confounders such as age, sex, body mass index (BMI), physical activity, or socioeconomic status were adjusted for in the analysis. These are important factors that could influence the results.

16. The logistic regression results indicate that glucose and insulin were the only predictors of insulin resistance, but the outcome is expected given that HOMA-IR is calculated from these two variables. We should acknowledge this limitation.

17. You stated, “The serum galectin-3 level was generally higher in participants with malaria and lowest in non-diabetic participants without malaria.” This sentence is confusing and should be rewritten for clarity.

18. There appears to be no stratification or matching between groups for key variables such as age, sex, BMI, or physical activity. These differences may confound the observed associations.

19. Galectin-3 is an inflammatory marker. However, no inflammatory or oxidative stress markers (e.g., CRP, IL-6, TNF-α) were included. Without these, it is difficult to contextualize galectin-3 elevation in relation to systemic inflammation or metabolic syndrome.

The study does not consider the impact of antimalarial or antidiabetic medications, like artemisinin derivatives or metformin, which can influence glucose metabolism and galectin-3 levels.

21. Insulin and glucose levels can vary throughout the day. Were samples collected during a standardized fasting window? If not, variability in sample timing could have affected HOMA-IR estimates.

Reviewer #2: The current study is relevant since diabetes is on the rise in the African region where malaria is highly endemic. Here the authors have made an attempt to find the role of galectin-3 with diabetes and malaria. They did not find concrete evidence the exact role galectin-3 irrespective of diabetes status, malaria and insulin resistance. Some suggestions:

1. Please mention whether the lifestyle is more responsible for diabetes and more so in the elderly participants.

2. Please mention the exact level of parasitaemia and species of malaria.

6. PLOS authors have the option to publish the peer review history of their article (what does this mean? ). If published, this will include your full peer review and any attached files.

**Do you want your identity to be public for this peer review?** For information about this choice, including consent withdrawal, please see our Privacy Policy .

Reviewer #1: **Yes: ** Majid Asgari

Reviewer #2: **Yes: ** Prof Susanta Kumar hosh

---

## [Author Response · Author response to Decision Letter 1]

29 May 2025

Appropriate responses have been provided for each of the comments by the reviewers in line with journal requirements and uploaded, please.

---

## [Decision Letter · Decision Letter 1]

24 Jun 2025

PONE-D-25-15329R1Galectin-3 did not associate with malaria-related insulin resistance in diabetic and non-diabetic respondents at a Ghanaian General HospitalPLOS ONE

Dear Dr. Acquah,

Thank you for submitting your manuscript to PLOS ONE. After careful consideration, we feel that it has merit but does not fully meet PLOS ONE’s publication criteria as it currently stands. Therefore, we invite you to submit a revised version of the manuscript that addresses the points raised during the review process.

Thank you very much for your attempt to address all comments. However, one of the best peer reviewer of your manuscript offered some additional comments in this revised version for your deliberation.

A rebuttal letter that responds to each point raised by the academic editor and reviewer(s). You should upload this letter as a separate file labeled 'Response to Reviewers'.A marked-up copy of your manuscript that highlights changes in GREEN made to the original version. You should upload this as a separate file labeled 'Revised Manuscript with Track Changes'.An unmarked version of your revised paper without tracked changes. You should upload this as a separate file labeled 'Manuscript'.

We look forward to receiving your revised manuscript.

Kind regards,

Mohammad Reza Mahmoodi, Ph.D.

Academic Editor

PLOS ONE

Reviewers' comments:

Reviewer's Responses to Questions

**Comments to the Author**

1. If the authors have adequately addressed your comments raised in a previous round of review and you feel that this manuscript is now acceptable for publication, you may indicate that here to bypass the “Comments to the Author” section, enter your conflict of interest statement in the “Confidential to Editor” section, and submit your "Accept" recommendation.

Reviewer #1: All comments have been addressed

Reviewer #2: All comments have been addressed

2. Is the manuscript technically sound, and do the data support the conclusions?

Reviewer #1: Yes

Reviewer #2: Yes

3. Has the statistical analysis been performed appropriately and rigorously? 

Reviewer #1: Yes

Reviewer #2: Yes

4. Have the authors made all data underlying the findings in their manuscript fully available?

Reviewer #1: Yes

Reviewer #2: Yes

5. Is the manuscript presented in an intelligible fashion and written in standard English?

Reviewer #1: Yes

Reviewer #2: Yes

6. Review Comments to the Author

Reviewer #1: Dear Authors,

Thank you very much for your thorough and comprehensive responses to my previous comments and suggestions. I appreciate the effort you've put into addressing the feedback.

I also took note of your comments regarding sentence length, my emotions and mood, and the background of English learners, and I have considered them while reviewing the revised version.

As mentioned earlier, I recommend seeking assistance from a native English speaker to help improve the readability and clarity of your manuscript.

Below are additional comments that reflect a more detailed and precise reading of the manuscript:

1. The term “chimeric” is used without any accompanying explanation of galectin-3’s structural characteristics. If this term is retained, please include a brief description of its chimeric structure; otherwise, consider omitting it to avoid confusion.

2. Please edit and make improvements. Figure 1: The current version is asymmetrical and might not be up to the target journal's visual standards.

3. The sentence “In addition, the region has the lowest prevalence of diabetes at 4.5%, but the highest proportion of undiagnosed cases at 53.6%” lacks a supporting reference. Please add a citation.

4. Causality and Study Design: While the study is cross-sectional, some phrases (e.g., “insulin resistance could be predicted by glucose and insulin”) imply causality. Such interpretations should be avoided in non-interventional studies. Use cautious language like “associated with.”

5. The statement “Galectin-3 was postulated to promote cerebral malaria…” refers to experimental work but does not distinguish between findings from animal models and human studies in Ghana. Please provide appropriate context and caveats.

6. Correct the spelling of “glectin-3” to “galectin-3” at P4.

7. Phrases like “averting severe illness” or “curbing the menace” are overly informal. Please consider more scientific alternatives, such as “mitigating disease severity” or “addressing disease burden.”

8. The paragraph introducing galectin-3 rapidly shifts between multiple diseases (T2DM, cancer, asthma, etc.). For coherence and relevance, consider narrowing the focus to metabolic and infectious diseases.

9. The introduction could be strengthened by acknowledging limitations or controversies in the existing literature—such as variability in galectin-3 levels due to comorbidities or unclear causative relationships in prior studies.

10. While the use of simple random sampling is appropriate, it should be acknowledged that hospital-based recruitment may introduce selection bias.

11. Please clarify whether malaria parasitemia (e.g., parasite density) was quantified. Severity stratification would add significant value beyond a binary malaria diagnosis.

12. Reiterate clearly that causality cannot be inferred due to the cross-sectional design.

13. While sample size calculation is addressed, please ensure that effect sizes are also reported in the results.

14. The phrase “Seven millimetres of venous blood” should be corrected to “Seven millilitres.”

15. The use of TMB and dual-wavelength readings (450 and 570 nm) is common, but a brief rationale should be included to enhance clarity for reproducibility.

16 Consider shortening overly technical assay descriptions. Emphasize essential elements such as the kit name, detection principle, sensitivity, and standard curve methodology.

17. While the limitations of a cross-sectional design are mentioned, causal phrases such as galectin-3 “worsening” hyperglycemia should be revised to more appropriate wording like “associated with.”

18. The explanation suggesting that lower parasite load in diabetics reflects immune dysfunction is speculative. This should be expanded or supported with relevant data (e.g., immune profiling or prior malaria exposure).

19. Avoid repetitive phrases like “in our context” or “in the current study” unless necessary for clarity.

Reviewer #2: The MS is OK now. The authors have addressed all the comments raised by the reviewers. Only I suggest to mention that falciparum is responsible for hypoglycemic conditions. Here how you will explain this in view of the present study.

7. PLOS authors have the option to publish the peer review history of their article (what does this mean? ). If published, this will include your full peer review and any attached files.

**Do you want your identity to be public for this peer review?** For information about this choice, including consent withdrawal, please see our Privacy Policy .

Reviewer #1: **Yes: ** Majid Asgari

Reviewer #2: **Yes: ** Dr Susanta Kumar Ghosh

---

## [Author Response · Author response to Decision Letter 2]

29 Jun 2025

This has been done and uploaded as the response to reviewers' document, please.

---

## [Decision Letter · Decision Letter 2]

28 Jul 2025

Galectin-3 did not associate with malaria-related insulin resistance in diabetic and non-diabetic respondents at a Ghanaian General Hospital

PONE-D-25-15329R2

Dear Dr. Acquah,

We’re pleased to inform you that your manuscript has been judged scientifically suitable for publication and will be formally accepted for publication once it meets all outstanding technical requirements.

Kind regards,

Mohammad Reza Mahmoodi, Ph.D.

Academic Editor

PLOS ONE

Additional Editor Comments (optional):

Reviewers' comments:

Reviewer's Responses to Questions

**Comments to the Author**

1. If the authors have adequately addressed your comments raised in a previous round of review and you feel that this manuscript is now acceptable for publication, you may indicate that here to bypass the “Comments to the Author” section, enter your conflict of interest statement in the “Confidential to Editor” section, and submit your "Accept" recommendation.

Reviewer #1: All comments have been addressed

2. Is the manuscript technically sound, and do the data support the conclusions?

Reviewer #1: Yes

3. Has the statistical analysis been performed appropriately and rigorously? 

Reviewer #1: Yes

4. Have the authors made all data underlying the findings in their manuscript fully available?

Reviewer #1: Yes

5. Is the manuscript presented in an intelligible fashion and written in standard English?

Reviewer #1: Yes

6. Review Comments to the Author

Reviewer #1: (No Response)

7. PLOS authors have the option to publish the peer review history of their article (what does this mean? ). If published, this will include your full peer review and any attached files.

**Do you want your identity to be public for this peer review?** For information about this choice, including consent withdrawal, please see our Privacy Policy .

Reviewer #1: No

---

## [Editor Report · Acceptance letter]

PONE-D-25-15329R2

PLOS ONE

Dear Dr. Acquah,

I'm pleased to inform you that your manuscript has been deemed suitable for publication in PLOS ONE. Congratulations! Your manuscript is now being handed over to our production team.

Kind regards,

on behalf of

Professor Mohammad Reza Mahmoodi

Academic Editor

PLOS ONE